# An Efficient Method of Calculating the Force and Torque in the Upsetting of Cylinders with Rotating Dies

Sergei Alexandrov [1] , Elena Lyamina [2] and Yeong-Maw Hwang [3],*

1. Department of Welding Engineering, Federal State Autonomous Educational Institution of Higher Education South Ural State University (National Research University), 76 Lenin Prospekt, 454080 Chelyabinsk, Russia; sergei_alexandrov@spartak.ru
2. Laboratory of Mechanics of Technological Processes, Ishlinsky Institute for Problems in Mechanics RAS, 119526 Moscow, Russia; lyamina@inbox.ru
3. Department of Mechanical and Electro-Mechanical Engineering, National Sun Yat-Sen University, 80424 Kaohsiung, Taiwan
* Correspondence: ymhwang@mail.nsysu.edu.tw

**Abstract:** The process of upsetting, with rotating dies, is used to reduce the force required to deform the workpiece and to receive more homogeneous deformation compared to the same process without the rotational motion of the dies. The upper bound method is an efficient tool for a quick estimate of process parameters. The accuracy of upper bound solutions depends on the chosen class of kinematically admissible velocity fields. The present paper provides an efficient method for choosing kinematically admissible velocity fields that satisfy some stress boundary conditions if the associated flow rule is considered. The method applies to the upsetting of cylinders. It is expected that it leads to accurate solutions if friction is high enough. Besides, the kinematically admissible velocity field accounts for a rigid region near the axis of symmetry. Such a region inevitably occurs in exact solutions because the friction stress must vanish at the axis of symmetry. The final expression for the upper bound, on a combination of the force and torque, involves two arbitrary parameters. These parameters are determined using the upper bound theorem. An example is provided to illustrate the method.

**Keywords:** upsetting; rotating dies; upper bound; rigid region; singularity





## 1. Introduction

The cylinder compression test is used for determining the flow stress of materials and friction between the tool and deforming material (for example, [1]). Teflon films significantly reduce friction such that the deformation is almost homogeneous [1]. Another method that allows for producing nearly uniform cylinder upsetting is Rastegaev's test [2]. However, in most cases, friction significantly affects the deformation process, including the force required to deform a cylinder and its lateral shape. There is a vast amount of literature on studying barreling during cylinder upsetting [3–7]. Different lubricants are usually used to study the effect of frictional conditions on barreling [8]. However, one can change the magnitude of the friction stress in the radial direction by using upsetting between rotating dies. The rotation of dies also reduces the force required to deform the specimen, which is of practical importance [9,10].

The upper bound method is a convenient and reliable tool for evaluating the load required to deform the workpiece in metal forming processes. The method is based on the upper bound theorem [11]. Several recent applications of the upper bound method to metal forming processes are available in [12–15]. In particular, numerous applications of this method to the cylinder compression test have appeared in the literature (for example, [8,16]). Paper [8] has presented an experimental/theoretical study on the effect of friction between the workpiece and tool on barreling of solid aluminum cylinders. The theoretical solution

has been based on the assumption that the bulge's shape, in a generic meridional plane, is a circular arc. It has been shown that the radius of curvature of bulge, measured experimentally, is in good agreement with its theoretical prediction for all three aspect ratios used in the experiment. Paper [16] has provided a collection of formulae for calculating lower and upper bounds on the pressure required to deform strips under plane strain conditions and axisymmetric disks. Two dimensionless parameters have been introduced, and the effect of these parameters on the pressure has been analyzed. Plastic anisotropy has been taken into account in [17], where the upsetting of hollow cylinders has been studied. It has been shown that this mechanical property may greatly affect the pressure required to deform the cylinder. These solutions have been extended to upsetting with rotating dies in [18–22]. In addition to accounting for the rotation of dies in the upper bound solutions above, paper [20] has compared the solutions found using the upper bound and slab methods. Paper [21] has extended this study to the upsetting of clad cylinders. An approach for using the upper bound theorem to analyze the continued compression of solid cylinders by rotating dies has been developed in [22]. A disadvantage of the solutions above is that no rigid region appears. On the other hand, such a region must appear in the exact solution because the regime of sticking friction inevitably occurs in the vicinity of the symmetry axis. The upper bound solution, found in [23], is based on a kinematically admissible velocity field that assumes the existence of a rigid region. This solution is for the upsetting of a cylinder with no twist. The present paper generalizes this solution to include the rotational motion of dies.

## 2. Statement of the Problem

A circular solid cylinder is upset between two rotating flat dies. The radius of the cylinder is $R$, and its height is $2H$ (Figure 1). The vertical velocity of each die is $U$, and its angular velocity is $\omega$. The force and torque applied to each die are $F$ and $T$, respectively. It is natural to use a cylindrical coordinate system $(r, \theta, z)$ such that its $z$-axis coincides with the axis of symmetry of the process, and the plane $z = 0$ coincides with the plane of symmetry of the process. Then, by virtue of symmetry, it is sufficient to consider the domain $z \geq 0$. Let $u_r$, $u_\theta$, and $u_z$ be the components of velocity in the radial, circumferential, and axial directions, respectively.

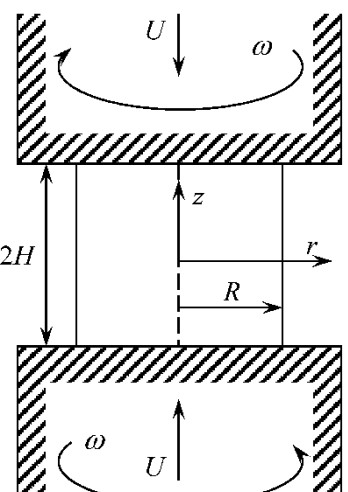

**Figure 1.** Schematic diagram of the process.

The velocity boundary conditions are

$$u_z = 0 \text{ for } z = 0, \tag{1}$$

$$u_z = -U \text{ for } z = H, \tag{2}$$

and

$$u_r = 0 \text{ for } r = 0. \tag{3}$$

The lateral surface of the cylinder is traction-free. By virtue of symmetry,

$$\sigma_{rz} = 0 \text{ and } \sigma_{\theta z} = 0 \text{ for } z = 0. \tag{4}$$

Here $\sigma_{rz}$ and $\sigma_{\theta z}$ are the shear stresses, referred to in the cylindrical coordinate system. Let $\tau_s$ be the shear yield stress. Its value is constant in the case of rigid perfectly plastic materials. The friction law at $z = H$ postulates that the friction stress, $\tau_f$, is equal to a constant fraction of the shear yield stress. However, the direction of the friction stress is unknown. Then,

$$\tau_f = m\tau_s \text{ for } z = H. \tag{5}$$

Here $0 \leq m \leq 1$. Equation (5) is valid at sliding.

The material of the cylinder obeys the von Mises yield criterion. In this case, the plastic work rate per unit volume is represented as

$$w = \sqrt{3}\tau_s \xi_{eq}. \tag{6}$$

Here $\xi_{eq}$ is the equivalent strain rate. Let $\xi_{rr}$, $\xi_{\theta\theta}$, $\xi_{zz}$, $\xi_{r\theta}$, $\xi_{\theta z}$ and $\xi_{zr}$ be the strain rate components referred to the cylindrical coordinate system. Then,

$$\xi_{eq} = \sqrt{\frac{2}{3}}\sqrt{\xi_{rr}^2 + \xi_{\theta\theta}^2 + \xi_{zz}^2 + 2\xi_{r\theta}^2 + 2\xi_{\theta z}^2 + 2\xi_{zr}^2}. \tag{7}$$

Taking into account that the solution is independent of $\theta$, one can express the strain rate components through the velocity components as

$$\xi_{rr} = \frac{\partial u_r}{\partial r}, \; \xi_{\theta\theta} = \frac{u_r}{r}, \; \xi_{zz} = \frac{\partial u_z}{\partial z},$$
$$\xi_{r\theta} = \frac{1}{2}\left(\frac{\partial u_\theta}{\partial r} - \frac{u_\theta}{r}\right), \; \xi_{\theta z} = \frac{1}{2}\frac{\partial u_\theta}{\partial z}, \; \xi_{zr} = \frac{1}{2}\left(\frac{\partial u_r}{\partial z} + \frac{\partial u_z}{\partial r}\right). \tag{8}$$

It is convenient to introduce the following dimensionless quantities:

$$\rho = \frac{r}{R}, \; \zeta = \frac{z}{H}, \; h = \frac{H}{R}, \; \chi = \frac{\omega R}{U}. \tag{9}$$

### 3. Upper Bound Solution

#### 3.1. Kinematically Admissible Velocity Field

Kinematically admissible velocity fields must satisfy the incompressibility equation and velocity boundary conditions. However, it is advantageous to choose a kinematically admissible velocity field that also satisfies some additional conditions, which follow from the behavior of the actual velocity field. These additional conditions depend on the boundary value problem. In the case under consideration, the symmetry of the process dictates that the radial velocity is an even function of $\zeta$. The other velocity components are odd functions of $\zeta$. This feature of the actual velocity field will be taken into account below. If $m = 1$ in (5), then [24]

$$\frac{\partial u_r}{\partial \zeta} = O\left(\frac{1}{\sqrt{1-\zeta}}\right) \text{ and } \frac{\partial u_\theta}{\partial \zeta} = O\left(\frac{1}{\sqrt{1-\zeta}}\right) \tag{10}$$

as $\zeta \to 1$. Several solutions have shown that it is advantageous to use (10) even if $m < 1$, though its value should be large enough [12,25]. Therefore, (10) will be taken into account below.

Analytical solutions for the plane strain compression of a layer between two parallel plates are available for various material models [26–28]. All these solutions show that the axial velocity is a linear function of $\zeta$. It is reasonable to assume this distribution of the

axial velocity occurs in the problem under consideration. Then, the boundary conditions (1) and (2) uniquely determine the axial velocity distribution. Taking into account (9), one gets

$$\frac{u_z}{U} = -\zeta. \tag{11}$$

The incompressibility equation follows from (8) in the form:

$$\xi_{rr} + \xi_{\theta\theta} + \xi_{zz} = \frac{\partial u_r}{\partial r} + \frac{u_r}{r} + \frac{\partial u_z}{\partial z} = 0 \tag{12}$$

Using (9) and (11), one transforms this equation to

$$\frac{\partial u_r}{\partial \rho} + \frac{u_r}{\rho} = \frac{U}{h}. \tag{13}$$

The general solution of this equation is

$$\frac{u_r}{U} = \frac{\rho}{2h} + \frac{f(\zeta)}{\rho}. \tag{14}$$

Here $f(\zeta)$ is an arbitrary function of $\zeta$. It is seen, from this solution, that condition (3) cannot be satisfied unless $f(\zeta) = 0$. On the other hand, it is not realistic to assume that $f(\zeta) = 0$ if $m \neq 0$. Equation (14) is compatible with condition (3) if there exists a rigid region containing the $z$-axis (Figure 2). The rigid region moves along the $z$-axis and rotates about this axis together with the die. Then, the velocity vector in this region is represented as

$$\mathbf{U_r} = \omega r \mathbf{j} - U \mathbf{k}. \tag{15}$$

Here $\mathbf{j}$ and $\mathbf{k}$ are the unit base vectors in the circumferential and axial directions, respectively. The unit base vector, in the radial direction, is denoted as $\mathbf{i}$.

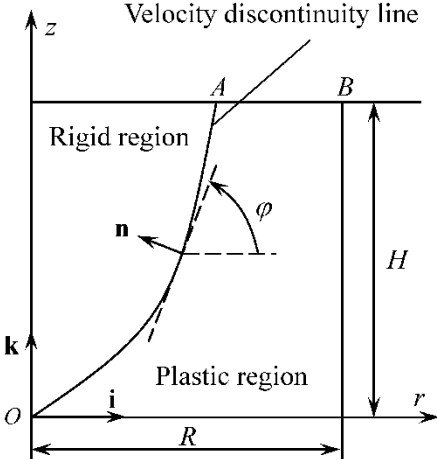

**Figure 2.** General structure of the kinematically admissible velocity field.

Since the circumferential velocity is an odd function of $\zeta$, it is reasonable to choose it in the form:

$$\frac{u_\theta}{U} = \rho \zeta g(\zeta). \tag{16}$$

Here $g(\zeta)$ is an arbitrary even function of $\zeta$.

Substituting (11), (14) and (16) into (8), one finds the strain rate components as

$$\xi_{rr} = \frac{U}{H}\left(\frac{1}{2} - \frac{hf(\zeta)}{\rho^2}\right), \quad \xi_{\theta\theta} = \frac{U}{H}\left(\frac{1}{2} + \frac{hf(\zeta)}{\rho^2}\right), \quad \xi_{zz} = -\frac{U}{H},$$
$$\xi_{\theta z} = \frac{U\rho}{2H}\left[g(\zeta) + \zeta\frac{dg}{d\zeta}\right], \quad \xi_{zr} = \frac{U}{2H\rho}\frac{df}{d\zeta}, \quad \xi_{r\theta} = 0. \tag{17}$$

Equations (7) and (17) combine to give

$$\xi_{eq} = \frac{U}{H}\sqrt{\frac{2}{3}}\sqrt{\frac{3}{2} + \frac{2h^2 f^2(\zeta)}{\rho^4} + \frac{\rho^2}{2}\left[g(\zeta) + \zeta\frac{dg}{d\zeta}\right]^2 + \frac{1}{2\rho^2}\left(\frac{df}{d\zeta}\right)^2}. \tag{18}$$

The velocity normal to the velocity discontinuity surface must be continuous. Let **n** be the unit vector normal to line *OA* (Figure 2). This line is generated by the intersection of the velocity discontinuity surface and a generic meridian plane. It is seen from the geometry of Figure 2 that

$$\mathbf{n} = -\mathbf{i}\sin\varphi + \mathbf{k}\cos\varphi. \tag{19}$$

Here $\varphi$ is the inclination of the tangent to line *OA* to the *r*-axis, measured anticlockwise. The velocity vector in the plastic region is

$$\mathbf{U_p} = u_r\mathbf{i} + u_\theta\mathbf{j} + u_z\mathbf{k}. \tag{20}$$

The condition of continuity of the normal velocity can be represented as $\mathbf{U_p} \cdot \mathbf{n} = \mathbf{U_r} \cdot \mathbf{n}$. Substituting (15), (19) and (20) into this equation, one gets

$$u_r\sin\varphi - u_z\cos\varphi = U\cos\varphi. \tag{21}$$

It is understood here that $u_r$ and $u_z$ are to be calculated at line *OA*. It follows from the geometry of Figure 2 and (9) that

$$\tan\varphi = \frac{dz}{dr} = h\frac{d\zeta}{d\rho}. \tag{22}$$

Equations (11), (14), (21) and (22) combine to give

$$\frac{d\rho}{d\zeta} = \frac{\rho^2 + 2hf(\zeta)}{2\rho(1-\zeta)}. \tag{23}$$

It is convenient to use the following substitution:

$$\eta = \rho^2. \tag{24}$$

Then, Equation (23) becomes

$$\frac{d\eta}{d\zeta} = \frac{\eta + 2hf(\zeta)}{1-\zeta}. \tag{25}$$

This is a linear differential equation. Its general solution is

$$\eta = \eta_{OA}(\zeta) = \frac{2h\left(\int\limits_1^\zeta f(\lambda)d\lambda + C\right)}{1-\zeta}. \tag{26}$$

It is seen from this solution that $\eta \to \infty$ as $\zeta \to 1$ unless $C = 0$. Therefore, lines *OA* and $\zeta = 1$ have a common point only if $C = 0$. In this case, Equation (26) becomes

$$\eta = \eta_{OA}(\zeta) = \frac{2h\int\limits_1^\zeta f(\lambda)d\lambda}{1-\zeta}. \tag{27}$$

This equation determines the shape of line *OA* if the function $f(\zeta)$ is prescribed. The right-hand side of (27) reduces to the expression 0/0 as $\zeta \to 1$. Using l'Hospital's rule, one gets

$$\eta_A = -2hf(1). \tag{28}$$

Here $\eta_A$ is the value of $\eta$ at point $A$ (Figure 2). Moreover, line $OA$ must contain the origin of the coordinate system. Then, it follows from (27) that

$$\int_0^1 f(\zeta)d\zeta = 0. \tag{29}$$

*3.2. Upper Bound Theorem*

In the case under consideration, the upper bound theorem reads

$$FU + T\omega \leq \Omega_V + \Omega_d + \Omega_f. \tag{30}$$

Here $\Omega_V$ is the plastic work rate in the plastic region, $\Omega_d$ is the plastic work rate at the velocity discontinuity surface, and $\Omega_f$ is the plastic work rate at the friction surface. Using (6), one determines $\Omega_V$ as

$$\Omega_V = 2\sqrt{3}\pi\tau_s \int_0^H \int_{r_{OA}(z)}^R \xi_{eq} r\,dr\,dz. \tag{31}$$

Here $r = r_{OA}(z)$ is the equation of the velocity discontinuity line $OA$ in the cylindrical coordinates. Using (9), (18) and (24), one transforms (31) to

$$\frac{\Omega_V}{\sqrt{3}\pi\tau_s R^2 U} = \sqrt{\frac{2}{3}} \int_0^1 \int_{\eta_{OA}(\zeta)}^1 \sqrt{\frac{3}{2} + \frac{2h^2 f^2(\zeta)}{\eta^2} + \frac{\eta}{2}\left[g(\zeta) + \zeta\frac{dg}{d\zeta}\right]^2 + \frac{1}{2\eta}\left(\frac{df}{d\zeta}\right)^2} \, d\eta\,d\zeta. \tag{32}$$

The plastic work rate, at the velocity discontinuity surface, is determined as

$$\Omega_d = 2\pi\tau_s \int [u_\tau]r\,dL. \tag{33}$$

Here $dL$ is the infinitesimal length element of line $OA$ and

$$[u_\tau] = \left|\mathbf{U_r} - \mathbf{U_p}\right| \tag{34}$$

where $\mathbf{U_r}$ and $\mathbf{U_p}$ are understood to be calculated at line $OA$. Using (15) and (20), one finds

$$[u_\tau] = \sqrt{u_r^2 + (u_\theta - \omega r)^2 + (u_z + U)^2}. \tag{35}$$

Eliminating the velocity vector components in the plastic region, employing (11), (14) and (16) yields

$$[u_\tau] = U\sqrt{\left[\frac{\sqrt{\eta_{OA}(\zeta)}}{2h} + \frac{f(\zeta)}{\sqrt{\eta_{OA}(\zeta)}}\right]^2 + \eta_{OA}(\zeta)[\zeta g(\zeta) - \chi]^2 + (1-\zeta)^2}. \tag{36}$$

In deriving this equation, (9) and (24) have been used.

By definition,

$$dL = \sqrt{(dr)^2 + (dz)^2} = \sqrt{\left(\frac{dr}{dz}\right)^2 + 1}\,dz. \tag{37}$$

The derivative $dr/dz$ can be eliminated employing (22) and (23). The resulting equation and (24) combine to give

$$dL = H\sqrt{\frac{[\eta_{OA}(\zeta) + 2hf(\zeta)]^2}{4h^2\eta_{OA}(\zeta)(1-\zeta)^2} + 1}\, d\zeta. \tag{38}$$

Substituting (36) and (38) into (33) leads to

$$\frac{\Omega_d}{\sqrt{3}\pi\tau_s R^2 U} = \frac{2h}{\sqrt{3}}\int_0^1 \sqrt{\frac{\left[\frac{\sqrt{\eta_{OA}(\zeta)}}{2h} + \frac{f(\zeta)}{\sqrt{\eta_{OA}(\zeta)}}\right]^2 + \eta_{OA}(\zeta)[\zeta g(\zeta) - \chi]^2 + (1-\zeta)^2 \times}{\sqrt{\frac{1}{4h^2}\left[\frac{\eta_{OA}(\zeta) + 2hf(\zeta)}{(1-\zeta)}\right]^2 + \eta_{OA}(\zeta)}}}\, d\zeta. \tag{39}$$

The plastic work rate at the friction surface involves the actual friction stress. The magnitude of the friction stress is given by (5). However, the direction of the friction stress vector is controlled by the actual velocity field, which is unknown. Therefore, $\Omega_f$ cannot be evaluated using a kinematically admissible velocity field. Let $\gamma$ be the angle between the actual velocity vector and a kinematically admissible velocity vector at any point of the friction surface. Here, both velocities are understood to be the velocities relative to the tool surface. Then,

$$\Omega_f = m\tau_s \int \sqrt{u_r^2 + (u_\theta - \omega r)^2}\, \cos\gamma\, dS_f. \tag{40}$$

Here $S_f$ is the friction surface. The velocity components are understood to be calculated at the friction surface. Since $\cos\gamma \leq 1$, it is seen from (40) that

$$\Omega_f \leq \Omega_f^{(a)} = m\tau_s \int \sqrt{u_r^2 + (u_\theta - \omega r)^2}\, dS_f. \tag{41}$$

Therefore, Equation (30) can be rewritten as

$$FU + T\omega \leq \Omega_V + \Omega_d + \Omega_f^{(a)}. \tag{42}$$

Substituting (14) and (16) into (41) and using (9), (24) and (28), one arrives at

$$\frac{\Omega_f^{(a)}}{\sqrt{3}\pi\tau_s R^2 U} = \frac{m}{2\sqrt{3}h}\int_{\eta_A}^1 \sqrt{\eta}\sqrt{\left[1 - \frac{\eta_A}{\eta}\right]^2 + 4h^2[g(1) - \chi]^2}\, d\eta. \tag{43}$$

Using (9), one can rewrite Equation (42) as

$$\Lambda_u = \frac{\Omega_V + \Omega_d + \Omega_f^{(a)}}{\sqrt{3}\pi\tau_s R^2 U} \tag{44}$$

where $\Lambda_u$ is the upper bound on the quantity $(F + \chi T/R)/\left(\sqrt{3}\pi\tau_s R^2\right)$. Substituting (32), (39) and (43) into (44), one can calculate $\Lambda_u$ if the functions $f(\zeta)$ and $g(\zeta)$ are prescribed.

### 3.3. Choice of the Functions $f(\zeta)$ and $g(\zeta)$

Since $f(\zeta)$ is an even function of $\zeta$, it follows, from (14), that one of the simplest choices for this function, satisfying (10), is

$$f(\zeta) = \mu_0 + \mu_1\sqrt{1 - \zeta^2} \tag{45}$$

where $\mu_0$ and $\mu_1$ are constant. It follows from (28) and (45) that

$$\mu_0 = -\frac{\eta_A}{2h}. \tag{46}$$

Substituting (45) into (29), one gets

$$\mu_0 = -\frac{\mu_1 \pi}{4}. \tag{47}$$

Equations (45)–(47) combine to give

$$f(\zeta) = -\frac{\eta_A}{2h}\left(1 - \frac{4}{\pi}\sqrt{1 - \zeta^2}\right). \tag{48}$$

Using (24), one can find from (14) and (48) that $u_r = 0$ at $\eta = \eta_A$. Then, it is reasonable to require that

$$u_\theta = \omega r \text{ at } \eta = \eta_A. \tag{49}$$

By analogy to (45), one chooses the function $g(\zeta)$ in the form

$$g(\zeta) = \nu_0 + \nu_1\sqrt{1 - \zeta^2} \tag{50}$$

where $\nu_0$ and $\nu_1$ are constant. Equations (9), (16), (49) and (50) combine to give $\nu_0 = \chi$. Then, Equation (50) becomes

$$g(\zeta) = \chi + \nu_1\sqrt{1 - \zeta^2}. \tag{51}$$

Substituting (48) and (51) into (32), (39) and (43) allows one to determine the right-hand side of (44) as a function of $\eta_A$ and $\nu_1$. A minimum value of this function gives the best upper bound $\Lambda_u$ based on the kinematically admissible velocity field chosen. Minimization should be performed numerically.

## 4. Numerical Example

The boundary value problem is classified by three dimensionless parameters, namely $h$, $m$, and $\chi$. Since the present paper emphasizes the effect of die rotation on the upsetting process, the numerical example below focuses on the effect of $\chi$ on $\Lambda_u$. It is assumed that $h = 1$ in all calculations. The right-hand side of (44) has been minimized with respect to $\eta_A$ and $\nu_1$ using the Wolfram Mathematica software. Figure 3 depicts the variation of $\Lambda_u/\Lambda_0$ with $\chi$ for four values of $m$ ($m = 0.7$, $m = 0.8$, $m = 0.9$, and $m = 1$). Here $\Lambda_0$ is the value of $\Lambda_u$ at $\chi = 0$. The difference between the curves is very small and is invisible in the figure. It is an advantage of this representation of the solution. In particular, one curve provides $\Lambda_u/\Lambda_0$ for any $m$ in the range $0.7 \leq m \leq 1$. It is then necessary to calculate $\Lambda_0$ for a specific value of $m$ to determine the dependence of $\Lambda_u$ on $\chi$. Table 1 presents the value of $\Lambda_0$ for several values of $m$.

The solution found is not accurate if $m < 0.7$. In particular, minimizing the right-hand side of (44) leads to very small values of $\eta_A$, which means that the rigid region is vanishing. On the other hand, a rigid region must exist in the exact solution. An accurate solution requires a kinematically admissible velocity field that permits a rigid region while not penetrating the through-thickness of the cylinder.

**Table 1.** Dependence of $\Lambda_u$ on $m$ at $\chi = 0$.

| $m$ | 1 | 0.9 | 0.8 | 0.7 |
|---|---|---|---|---|
| $\Lambda_0$ | 1.171 | 1.168 | 1.164 | 1.156 |

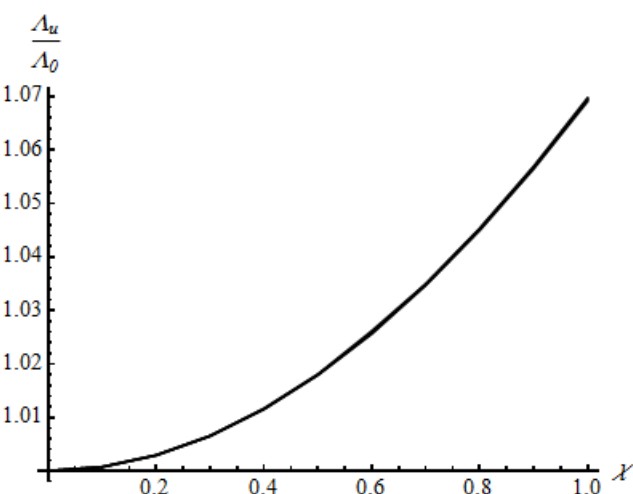

**Figure 3.** Variation of $\Lambda_u/\Lambda_0$ with $\chi$ at $h = 1$. This curve accurately represents $\Lambda_u/\Lambda_0$ if the friction factor is within the range $0.7 \le m \le 1$.

## 5. Conclusions

An upper bound solution, for the upsetting of a cylinder between rotating dies, has been proposed. The solution accounts for the singular behavior of the real velocity field near maximum friction surfaces. From this solution, the following conclusions can be drawn:

1.  The upper bound theorem does not immediately apply because the direction of the friction stress in the exact solution is unknown. For this reason, $\Omega_f$ in (30) has been replaced with $\Omega_f^{(a)}$ resulting in (44).
2.  An advantage of the solution is that the single curve, shown in Figure 3, accurately represents the variation of $\Lambda_u/\Lambda_0$ with $m$ in the range $0.7 \le m \le 1$.
3.  The solution is not appropriate if $m < 0.7$ (approximately). It predicts a vanishing rigid region in this range of the friction factor. In this case, a kinematically admissible velocity field that permits a rigid region, while not penetrating the through-thickness of the cylinder, is required. To the best of authors' knowledge, no such field has been proposed, even for the upsetting between non-rotating dies. This will be the subject of further investigation.

High-pressure torsion is a widely used severe plastic deformation process [28–32]. The solution given in the present paper can be adapted for this process. The cylinder cannot expand radially in the high-pressure torsion process. Then, the incompressibility equation demands that the axial velocity vanishes. The kinematically admissible velocity, proposed in the present paper, is applicable if the radial and axial velocities vanish. Then, the circumferential velocity is the only non-zero velocity component. Equation (16), in which $U$ should be replaced with $\omega R$, is valid. Another difference between the high-pressure torsion process and the process considered in the present paper is that the lateral surface of the cylinder is not traction-free in the former. However, because the radial velocity vanishes on this surface, it is only necessary to add, to the right-hand side of (30), the plastic work due to friction. The latter can be calculated in the same manner as (41).

**Author Contributions:** Formal analysis, S.A. and E.L.; conceptualization, S.A. and Y.-M.H.; writing—original draft, Y.-M.H. and E.L. All authors have read and agreed to the published version of the manuscript.

**Funding:** This research received funding from the Ministry of Science and Technology of the Republic of China under the grants MOST 108-2923-E-110-002-MY3 and MOST 109-2221-E-110-001-MY3.

**Institutional Review Board Statement:** Not applicable.

**Informed Consent Statement:** Not applicable.

**Data Availability Statement:** Not applicable.

**Conflicts of Interest:** The authors declare no conflict of interest.

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
