# Peer review of "An Efficient Method of Calculating the Force and Torque in the Upsetting of Cylinders with Rotating Dies"

_processes, doi:10.3390/pr9101845_

Round 1

Reviewer 1 Report

In the article it is necessary to discuss in more detail the previous conclusions of other authors dealing with this issue. The introduction is quite brief. The quality of the images and their informative value could be at a higher level and it would be necessary to redraw them and color-code the examined parameters. The mathematical apparatus of process description is at a good level and fully describes the whole analyzed process. The problem is extensive, so it may have been good to analyze some processes in more detail, but this is solely a matter for the authors. I agree with the presented conclusions. The paper can be published after minor modifications, because it brings new knowledge and broadens the scientific horizon.

Author Response

We have extended the Introduction. The corrections appear between lines 45 and 61 of the revised manuscript. They are shown in red.

As to the technical quality of images, we will consult the journal office. We always use this software to prepare images. Usually, the quality of images is acceptable. In any case, we will inform the journal office and resolve this issue.

Reviewer 2 Report

Dear Authors, 

The research proposes an original field for the solution of the upset of a cylinder between two rotating dies. For the sake of clarity, the authors should better explain the assumption made in the axial velocity, namely, Eq. (11). Also, it would be beneficial to the reader to widen the invisible difference in Figure 4. Lastly, the authors should highlight that the proposed method could be applied to severe plastic deformation processes involving compression and twisting.

Yours sincerely,

Reviewer

Author Response

The choice of the axial velocity in (11) is motivated by several available analytic solutions, as explained in the manuscript above this equation. The boundary conditions (1) and (2) also suggest this form. We have emphasized the latter in the revised manuscript.

The difference in the ordinates of the lines in Fig.4 is much less than 1%. It is necessary to enlarge the figure 100 times to make this difference visible.

Some modifications are required to apply the method to HPT. However, the reviewer is right that it is in general possible. We have modified the Conclusions to show that the method applies to HPT. This additional text is shown in red.